# An RNAi-Mediated Reduction in Transcription Factor Nrf-2 Blocks the Positive Effects of Dimethyl Fumarate on Metabolic Stress in Alzheimer’s Disease

**DOI:** 10.3390/ijms241411303

**Published:** 2023-07-11

**Authors:** Marika Lanza, Rossella Basilotta, Salvatore Cuzzocrea, Maria Bulzomì, Salvatore Oddo, Giovanna Casili, Emanuela Esposito

**Affiliations:** Department of Chemical, Biological, Pharmaceutical and Environmental Sciences, University of Messina, Viale F. Stagno d’Alcontres 31, 98166 Messina, Italy; rossella.basilotta@unime.it (R.B.); salvator@unime.it (S.C.); maria.bulzomi@unime.it (M.B.); salvatore.oddo@unime.it (S.O.); eesposito@unime.it (E.E.)

**Keywords:** metabolic stress, neurodegeneration, Alzheimer’s disease, brain, oxidative stress, neuroinflammation

## Abstract

The prevalence of obesity is rapidly rising around the world, and this will have a significant impact on our society as it is believed to be one of the leading causes of death. One of the main causes of these occurrences is added sugar consumption, which is associated with a higher risk of obesity, heart disease, diabetes, and brain illnesses such as Alzheimer’s disease (AD). To this purpose, excess sugar might worsen oxidative damage and brain inflammation: two neuropathological signs of AD. Dimethyl fumarate (DMF) is an orally accessible methyl ester of fumaric acid with putative neuroprotective and immunomodulatory properties. In addition, DMF stimulates the nuclear factor erythroid 2-related factor 2 (Nrf-2), a key regulator of the antioxidant response mechanism in cells. The aim of the current study was to assess the potential therapeutic benefits of DMF in an in vitro model of metabolic stress induced by high and low sugar levels. We discovered that DMF reversed the negative impacts of high and low glucose exposure on the viability and oxidative stress of SH-SY5Y cells. Mechanistically, DMF’s actions were mediated by Nrf-2. To this end, we discovered that DMF boosted the expression of the Nrf-2-regulated genes heme-oxygenase-1 (HO1) and manganese superoxide dismutase (MnSOD). More importantly, we found that inhibiting Nrf-2 expression prevented DMF’s positive effects. Our combined findings suggest that DMF may be a valuable support for treatments for metabolic diseases.

## 1. Introduction

Brain metabolic dysfunction alters brain activity in several neurological disorders, including Alzheimer’s disease (AD) [1]. Increased consumption of added sugar is a major contributor to common diseases associated with metabolic dysfunction such as obesity and diabetes mellitus, both of which are common AD risk factors [2]. To this end, high sugar intake causes neuroinflammation, oxidative stress, synapse loss, and degeneration of specific neuronal populations [3]. Consistent with these observations, high sugar intake is linked to hippocampal-dependent cognitive deficits in rodents and mild cognitive impairments in people [4,5,6,7]. Oxidative stress occurs mainly as a result of the overproduction of oxygen free radicals by mitochondria; in this context, the detoxification process is mediated by the mitochondrial antioxidant enzyme manganese superoxide dismutase (Mn-SOD) [8], which is significantly increased in AD patients [9]. Glutathione (GSH) and heme oxygenase 1 (HO-1) are at the center of another crucial antioxidant mechanism for the nervous system under stress conditions. Converging evidence indicates that alterations in GSH and HO-1 function are associated with brain inflammation, neurodegeneration, mild cognitive impairments, and AD [10,11]. The transcription factor nuclear factor (erythroid-derived 2)-like 2 (Nrf-2) plays a vital defensive role in orchestrating the antioxidant response in the brain. Nrf-2 activation promotes the expression of several antioxidant enzymes that exert cytoprotective effects against oxidative damage and mitochondrial impairment [12]. Interestingly, Nrf-2 signaling, activated by reactive oxygen species (ROS), induces the expression of antioxidant enzymes such HO-1, catalase, and Mn-SOD, which neutralize ROS, protecting cells against oxidative stress damage [13]. Consistent with these observations, a lack of the Nrf-2 gene exacerbates cognitive deficits in a mouse model of AD [14]. In recent years, new Nrf-2 drug activators have been synthesized, including dimethyl fumarate (DMF), a fumaric acid ester with antioxidant properties. Clinically, DMF is used as a treatment for multiple sclerosis [15]. DMF exerts its protective effects by activating the Nrf-2 antioxidant pathway and its anti-inflammatory effects by inhibiting NF-κB activity and reducing the expression of pro-inflammatory mediators [16]. In this study, we investigated the effects of DMF on the modulation of neuronal metabolic stress induced by high or low sucrose exposure. We report that DMF ameliorates metabolic-stress-induced cell death by an Nrf-2-mediated mechanism. Considering the role of Nrf-2 in cancerous cells, particularly the responsibility of Nrf-2 activation in the development of chemoresistance inactivating drug-mediated oxidative stress that normally leads cancer cells to death [17,18,19], these results may also have important clinical implications in cancer research.

## 2. Results and Discussion

### 2.1. DMF Increases Cell Viability in SHSY5Y Cells under Metabolic Stress

To assess the effects of glucose levels on cell viability, we cultured SH-SY5Y cells in the absence of glucose (GD) or in the presence of high glucose (17.5 mM; HG). We found that cell viability was markedly reduced in cells cultured in GD media compared to the control group (Figure 1A). We next sought to determine the effects of various concentrations of DMF, administered for 24 h, on metabolic stress-induced cell death. We found that administration of 0.1 mM DMF for 24 h ameliorated the effects of GD on cell viability. Indeed, we found a ~30% increase in cell viability in the GD + 0.1 DMF group compared to the GD group (Figure 1A). The protective effects of DMF were dose dependent. To this end, 1 mM DMF led to a ~10% increase in cell viability compared to the GD group, while 10 mM DMF had no significant effects compared to the GD group (Figure 1A).

In contrast, cells grown in HG media showed a ~59% decrease in cell viability compared to the control group (Figure 1B). Notably, 0.1 and 1 mM DMF significantly rescued HG-mediated cell death by 46% and 22%, respectively (Figure 1B), while 10 mM DMF had no effect (Figure 1B). Given these results, we decided to continue investigating only 0.1 and 1 mM DMF concentrations in subsequent studies.

### 2.2. DMF Modulates Oxidative Stress Induced by Glucose Deprivation

Impaired energy metabolism in neurons and subsequent oxidative stress are integral to many central nervous system diseases [20]. The overproduction of reactive oxygen species (ROS), along with the imbalance of the antioxidant enzyme systems of the body destroys cellular structures, lipids, proteins, DNA, and RNA [21]. To determine whether the antioxidant response system was involved in the effects of DMF, we evaluated the effect of DMF on the expression of MnSOD, HO-1, catalase and Nrf-2 by Western blot analysis (Figure 2A). We found that the steady-state levels of HO-1 were markedly higher in the GD group (Figure 2B). This increase was significantly mitigated by 0.1 and 1 mM DMF (*p* < 0.001; Figure 2B).

To further dissect the effects of GD and DMF on oxidative damage, we measured the levels of MnSOD. We found that GD significantly reduced the steady-state levels of MnSOD. Treatment with 0.1 mM DMF not only rescued the effects of GD but further increased MnSOD by about fourfold over the control group (*p* < 0.001; Figure 2C). In contrast, 1 mM DMF did not rescue the changes in MnSOD levels induced by GD (Figure 2C). Further studies are needed to understand the overshot in MnSOD levels induced by 0.1 mM DMF.

Furthermore, we found that GD significantly reduced the steady-state levels of catalase and Nrf-2 and that 0.1 mM and 1 mM DMF prevented these changes in a concentration-dependent manner (Figure 2D,E).

To better understand the mechanisms underlying the changes in oxidative stress induced by GD and DMF, we measured the levels of GSH and reactive oxygen species modulator 1 (ROMO1) by ELISA. ROMO1 is a protein in the inner space of mitochondria involved in reducing ROS production [22]. We found that GD significantly reduced GSH concentration (Figure 2F). These changes were rescued by DMF treatment in a dose-dependent manner (*p* < 0.001; Figure 2F). Furthermore, while GD significantly increased ROMO-1 levels, 0.1 mM (*p* < 0.01) and 1 mM (*p* < 0.001) DMF mitigated and completely rescued the increase in ROMO-1 levels, respectively (Figure 2G).

### 2.3. DMF Modulates Oxidative Stress Induced by High Glucose Levels

The same parameters were also evaluated in conditions of metabolic stress induced by high glucose levels (Figure 3A). We found that the steady-state levels of MnSOD were significantly reduced in the HG group compared to the CTL group (Figure 3C). Notably, DMF not only prevented the HG-induced changes in MnSOD levels, but it further increased them over the CTL group (Figure 3C). Similar results were obtained when we measured HO-1 levels. Specifically, we found that the steady-state levels of HO-1 were reduced in the HG group. While 0.1 mM DMF had no effects on this reduction, 1 mM DMF prevented the high glucose-induced changes in HO-1 levels (Figure 3B). In addition, we found that HG significantly reduced the levels of catalase (Figure 3D) and Nrf-2 (Figure 3E) and that both 0.1 mM and 1 mM DMF prevented these changes. To further evaluate the effects of HG on the cells’ antioxidant response, we measured GSH and ROMO1 by ELISA. We found that GSH levels were reduced by high sucrose and 0.1 DMF prevented these changes (Figure 3F). Unexpectedly, 1 mM DMF exacerbated the effects of high glucose on GSH levels (Figure 3F). Further studies are needed to better understand these changes. When we measured ROMO1 levels, we found that high sucrose significantly decreased its steady-state levels while 0.1 and 1 mM DMF prevented these changes (Figure 3G). Together, these results indicate that overall, DMF mitigates changes in the antioxidant response induced by metabolic distress.

### 2.4. DMF Reduces Neuroinflammation Induced by Metabolic Distress via an NF-κB-Mediated Mechanism

Altered energy metabolism in neurons is an integral part of many diseases of the central nervous system. Strong evidence indicates that metabolic stress increases neuroinflammation [23]. NF-κB, a well-studied mediator of inflammation and immunity, unites metabolic responses to inflammation [24]. To this end, strong evidence supports a deep correlation between oxidative stress and antioxidant decline with the concomitant development of neuroinflammation as factors underlying the pathogenesis of many central nervous system diseases including AD, PD and HD [25]. To assess whether metabolic distress alters markers of inflammation in our system, we first focused on NF-κB, a transcription factor that regulates the expression of many genes involved in the immune response [26]. We found that glucose deprivation significantly increased nuclear NF-κB levels (Figure 4A,B). Notably, this increase was prevented by 0.1 mM DMF, while 1 mM DMF decreased the steady-state levels of nuclear NF-κB below the CTL group (Figure 4A,B). To better understand the mechanisms linking DMF to NF-κB, we focused on IκBα and optineurin (OPTN), two key negative regulators of NF-κB; they suppress the translocation of NF-κB from the cytosol to the nucleus thereby blocking the transcriptional activity of NF-κB. Indeed, overexpression of OPTN suppresses neuroinflammation during the development and progression of diseases such as AD [27]. We found that IκB-α levels were significantly decreased by glucose deprivation. While 0.1 mM DMF did not counteract the effects of glucose deprivation on IκB-α, 1 mM DMF did. Indeed, the steady-state levels of IκB-α in cells treated with 1 mM DMF were not statistically significant compared to those of the CTL group (Figure 4C,D). In addition, we found that GD significantly decreased OPTN levels. However, this decrease was reversed in a dose-dependent manner by DMF treatment (Figure 4C,E). Taken together, these data suggest that GD activates NF-κB by reducing the expression of two inhibitory proteins, OPTN and IκB-α. In addition, the strong effects of 1 mM DMF on IκB-α and OPTN may account for the significant reduction in nuclear NF-κB levels after 1 mM DMF treatment.

To better understand the link between metabolic stress and neuroinflammation, we measured the levels of NF-κB in cells raised in high-glucose media with and without DMF. We found that NF-κB expression was increased in cells exposed to high glucose (Figure 5A,B). Administration of 0.1 and 1 mM DMF prevented the increase in NF-κB expression due to high sucrose (Figure 5A,B). Mechanistically, it appeared that the effects on NF-κB were mediated by changes in IκB-α and OPTN. To this end, we found that the steady-state levels of these two proteins were significantly reduced in the HG group, but these effects were, at least partially, reversed by DMF (Figure 5C–E).

### 2.5. DMF Prevented the p53 Changes Induced by Metabolic Stress

The p53 family of transcriptional activators constitutes a unique signaling network that mediates key cellular responses, including cell proliferation and stress responses, through the regulation of energy metabolism and oxidative stress [28,29]. To determine if p53 might be involved in the DMF-mediated improvements following metabolic stress, we stained cells with a p53-specific antibody. We found that p53 immunoreactivity was markedly reduced in cells exposed to glucose deprivation (Figure 6D–F,N). Notably, DMF prevented the glucose deprivation-induced changes in p53 levels in a dose-dependent manner (Figure 6G–L,N). We obtained similar results in cells exposed to high glucose (Figure 7). Specifically, we found that high glucose decreased p53 immunoreactivity, which was restored by DMF in a dose-dependent manner (Figure 7G–L,N). These data were also confirmed by Western blot analysis, which showed that metabolic stress reduced the steady-state levels of p53, while DMF mitigated these effects (Figure 6M and Figure 7M). Taken together, these data further highlight the role of DMF in increasing the neuronal defense system.

### 2.6. Nrf-2 Silencing Abolished DMF Protective Effect on Metabolic Stress

We have provided compelling evidence suggesting that DMF protects cells from metabolic stress by mitigating oxidative stress and inflammation. To better understand these mechanisms, we focused on Nrf-2, a master regulator of cells’ antioxidant and immune responses [30]. To do so, we employed siRNA technology to knock down the levels of Nrf-2 in SH-SY5Y cells and obtained a 40% reduction in Nrf-2 mRNA levels (Figure 8A). To determine if the effects of DMF were mediated by Nrf-2, we exposed cells to GD and HG after Nrf-2 knockdown. Notably, we found that after Nrf-2 knockdown, DMF did not prevent cell death induced by GD and HG (Figure 8B). In addition, we found that Nrf-2 knockdown significantly reduced HO-1 and MnSOD levels compared to CTR group (Figure 8D–F). Notably, treatment with 0.1 and 1 mM DMF was unable to increase the levels of these proteins in Nrf-2-knockdown cells. These data suggest that DMF, under metabolic stress, induces Nrf-2 expression, which in turn mounts an antioxidant and inflammatory response.

## 3. Materials and Methods

### 3.1. Materials

The SH-SY5Y cell line was purchased from the Shanghai Institutes for Biological Sciences (Shanghai, China). Dulbecco’s Modified Eagle Medium (DMEM)/F12 (including 17.5 mM glucose) and fetal bovine serum were purchased from Gibco Company (Grand Island, NY, USA). All chemicals were of the highest commercial grade available. All stock solutions were prepared in non-pyrogenic saline (0.9% NaCl; Baxter Healthcare Ltd., Thetford, Norfolk, UK) or 10% ethanol (Sigma-Aldrich, St. Louis, MO, USA).

### 3.2. Cell Culture

Human cell line SH-SY5Y was cultured in Dulbecco’s Modified Eagle Medium (DMEM)/F12 (Life Technologies, Gibco Company, Grand Island, NY, USA) supplemented with 10% fetal bovine serum (FBS, Life Technologies, Gibco^®^; Carlsbad, CA, USA), 100 U/mL of penicillin and 100 μg/mL of streptomycin. The cells were maintained in incubators at 37 °C with 5% CO_2_.

### 3.3. Cell Viability (MTT Assay)

SH-SY5Y cell viability was assessed using a mitochondria-dependent live cell dye (tetrazolium dye; MTT) (M5655; Sigma-Aldrich). Cells were plated on 96-well plates at a density of 4 × 10^4^ cells/well to a final volume of 150 μL. After 24 h, cells were treated with DMF (Sigma-Aldrich) for 24 h at concentrations of 0.1 mM, 1 mM, and 10 mM dissolved in basal medium. After 24 h the cells were incubated at 37 °C with MTT (0.2 mg/mL) for 1 h, the medium was removed by aspiration and cells were lysed with 100 μL DMSO. The extent of the reduction in MTT to formazan was quantified by measuring the optical density at 540 nm (OD540) with a microplate reader as previously described [22].

### 3.4. Western Blot Analysis

Western blot analysis was performed as previously described [31]. For cell lysates, SH-SY5Y cells were washed twice with ice-cold phosphate buffered saline (PBS), collected and resuspended in lysis buffer containing 20 mM Tris-HCl pH 7.5, 10 mM NaF, 150 μL of NaCl, 1% Nonidet P-40 and a protease cocktail of inhibitors (Catalog Number. 11836153001; Roche, Switzerland). After 40 min, cell lysates were centrifuged at 12,000 rpm for 15 min at 4 °C. Protein concentration was estimated using the Bio-Rad protein assay (Bio-Rad Laboratories, Hercules, CA, USA) using bovine serum albumin as a standard. The samples were then heated to 95 °C for 5 min and equal amounts of proteins were separated by 10–15% sodium dodecyl sulfate-polyacrylamide gel electrophoresis (SDS-PAGE) and transferred to a membrane of polyvinylidene difluoride (PVDF) (Immobilon-P, catalog number 88018; Thermo Fisher Scientific, Waltham, MA, USA). The following primary antibodies were used (Table 1): anti-HO-1 (1:500; sc-10789; Santa Cruz Biotechnology, St. Louis, MO, USA), anti-MnSOD (1:500; 06-984; Merck Millipore, Darmstadt, Germany), anti-catalase (1:500; sc-271803 Santa Cruz Biotechnology), anti-Nrf-2 (1:500; sc-365949 Santa Cruz Biotechnology), anti-p53 (1:500; sc-126 Santa Cruz Biotechnology), anti-IκB-α (1:500; sc-1643 Santa Cruz Biotechnology), anti-NF-κB (1:500; sc-8008; Santa Cruz Biotechnology), and anti-optineurin (1:500; sc-166578; Santa Cruz Biotechnology). The antibody dilutions were made in PBS with 5% skimmed milk powder and 0.1% Tween-20 (PMT) and the membranes were incubated overnight at 4 °C. The membranes were then incubated with a secondary antibody (1:2000; Jackson ImmunoResearch, West Grove, PA, USA) for 1 h at room temperature. To ensure that gels were loaded with equal amounts of protein lysate, they were also incubated with β-actin antibody (cytosolic fraction 1:1000; sc-47778; Santa Cruz Biotechnology) or laminin A/C (nuclear fraction 1:500; sc-376248; Santa Cruz Biotechnology). The signals were detected with an enhanced chemiluminescence (ECL) detection system mixture (Thermo Fisher, Waltham, MA, USA).

### 3.5. Enzyme Immunosorbent Assay (ELISA) for ROMO-1 and GSH

To evaluate the involvement of oxidative stress and the antioxidant response, the levels of ROMO-1 (human ROMO-1 ELISA kit Aviva System Biology cat# OKEH01371) and GSH (Cloud-Clone Corp. CEA294Ge) were measured in cell lysates by ELISA, according to the manufacturer’s instructions. Briefly, 50 µL of standards and cell lysates were added to the appropriate wells with 50 µL Detection Reagent A and incubated for 1 h at 37° C. The solution was then discarded, and three washes were performed with 1× wash solution. Then, 100 µL of 1× Detection Reagent B was added to each well and the plate was incubated for 30 min at room temperature. After repeated washings, 90 µL of substrate solution was added to each well and the plate was incubated for another 10–20 min. Finally, 50 µL of blocking solution was added to each well and the absorbance was read immediately using a microplate reader at 450 nm.

### 3.6. Immunofluorescence Analysis

Immunofluorescence analysis was performed as previously described [32]. The VSMCs on the coverslips were rinsed in phosphate-buffered saline (PBS: 0.15 M NaCl, 10 mM Na_2_HPO_4_, 3 mM NaN, pH 7.4), permeabilized in 0.2% Triton X-100 and blocked with 10% goat serum. Cells were incubated with an antibody against p53 (1:100, sc-126; Santa Cruz Biotechnology) overnight, followed by incubation with ITC-conjugated anti-mouse Alexa Fluor-488 antibody (1:2000 *v*/*v* Molecular Probes, Waltham, MA, USA) for 1 h at 37 °C. Sections were washed and 2 μg/mL 4′,6′-diamidino-2-phenylindole (DAPI; Hoechst, Frankfurt; Germany) was added for nuclear staining. Sections were observed and photographed at 40X magnification using a Leica DM2000 microscope.

### 3.7. Reverse Transcriptase PCR

Total RNA (2 μg) isolated from SH-SY5Y (4.5  ×  10^5^ cells on a 6 cm dish) was reverse transcribed, and synthesized cDNA was used as a template for PCR. RT-PCR was performed on a T100 Thermal Cycler (Bio-Rad, Hercules, CA, USA) with Taq polymerase (Life Technologies). cDNAs underwent 30 cycles for Nrf-2 and GAPDH using the following primers: Nrf-2 forward: TACTCCCAGGTTGCCCACA; Nrf-2 reverse: CATCTACAAACGGGAATGTCTGC. GAPDH forward: AATGACCCCTTCATTGAC, GAPDH reverse: TCCACGACGTACTCAGCGC. The PCR cycle used was as follows: 94 °C for 1 min, 52 °C for 45 s, and 72 °C for 55 s for a total of 21 cycles. After amplification, 10 μL of RT-PCR product was separated in 1.5% agarose gel electrophoresis in Tris/Borate/EDTA (0.089 M Tris-base, 0.089 M boric acid and 0.002 M EDTA). Nrf-2 levels were calculated as fold change relative to control, after normalization to GAPDH.

### 3.8. Small Interfering RNA Transfection

Cells were transfected with 100 nM siRNA against Nrf-2 (siRNA ID: #290473, Invitrogen by Thermo Fisher Scientific, Waltham, MA, USA) or 100 nM control siRNA (#4390843, Invitrogen by Thermo Fisher Scientific, Waltham, MA, USA) for 48 h using Lipofectamine transfection reagent (#18324-020, Invitrogen by Thermo Fisher Scientific, Waltham, MA USA) as previously described [22].

## 4. Conclusions

The data obtained in this study showed DMF modulation of metabolic stress condition, providing further evidence of its neuroprotective role. Although further studies are needed to elucidate the mechanisms underlying the reduction in oxidative damage induced by altered glucose metabolism, our data suggest that DMF may be a valid therapeutic compound in the management of metabolic stress and neurodegenerative diseases such as AD.

## Figures and Tables

**Figure 1 ijms-24-11303-f001:**
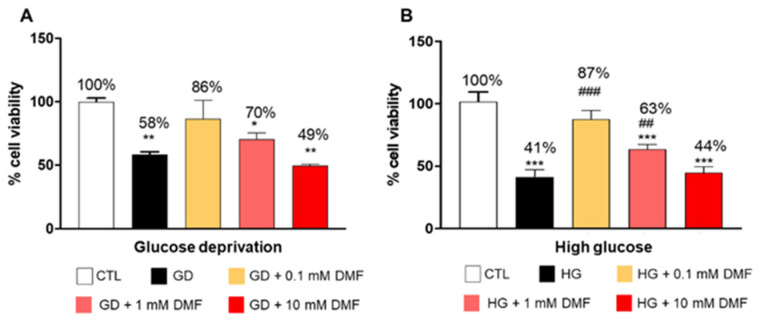
Effect of DMF on SHSY5Y cell viability. In cells grown in the absence of glucose, 0.1 and 1 mM DMF treatment for 24 h significantly increased cell viability compared to the GD group (**A**). In high glucose conditions, 0.1 mM DMF significantly increased SHSY5Y cell viability by 87% and 1 mM DMF by 63% compared to the HG group (**B**). The treatment with 10 mM DMF showed significant cytotoxicity in both conditions. Data are representative of at least three independent experiments (n = 3). Statistical analysis was performed by one-way ANOVA, followed by multiple comparisons performed by Bonferroni’s test. * *p* < 0.05 vs. CTR; ** *p* < 0.01 vs. CTR; *** *p* < 0.001 vs. CTR; ## *p* < 0.01 vs. GD/HG; ### *p* < 0.001 vs. GD/HG.

**Figure 2 ijms-24-11303-f002:**
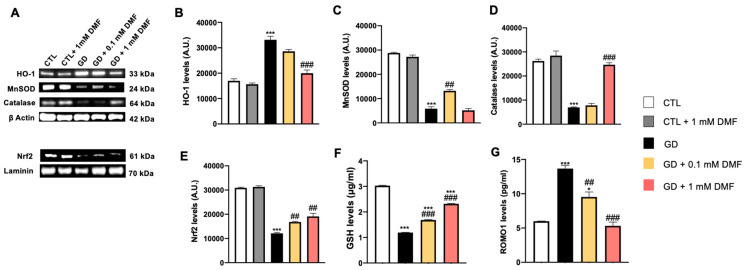
Effect of DMF on oxidative stress on SH-SY5Y cells after glucose deprivation. (**A**) Representative Western blots of proteins extracted from SH-SY5Y cells treated as indicated above the blots. Blots were probed with the indicated antibodies. β-actin and laminin were used as a loading controls. (**B**) Quantitative analyses of the blots revealed a significant increase in HO-1 levels in the GD group compared to the CTL group. In contrast, 0.1 and 1 mM DMF mitigated the GD-induced increase in HO-1 levels. *** *p* < 0.001 vs. CTR; ### *p* < 0.001 vs. GD. (**C**) Quantitative analyses of the blots revealed a significant decrease in MnSOD levels following glucose deprivation. A concentration of 0.1 mM but not 1 mM DMF significantly increased the expression of MnSOD following glucose deprivation. *** *p* < 0.001 vs. CTR; ## *p* < 0.01 vs. GD. (**D**) Quantitative analyses of the blots revealed a significant decrease in catalase levels following glucose deprivation. A concentration of 1 mM DMF significantly increased the expression of catalase following glucose deprivation. *** *p* < 0.001 vs. CTR; ### *p* < 0.001 vs. GD. (**E**) Quantitative analyses of the blots revealed a significant decrease in Nrf-2 levels following glucose deprivation. A concentration of 0.1 mM and 1 mM DMF significantly increased the expression of Nrf-2 following glucose deprivation. *** *p* < 0.001 vs. CTR; ## *p* < 0.01 vs. GD. (**F**,**G**) GSH and ROMO1 levels in culture media were measured by ELISA. GSH levels were significantly reduced by glucose deprivation. DMF significantly mitigated such a decrease. Glucose deprivation significantly increased ROMO1 levels; this increase was attenuated in a dose-dependent manner by DMF. In panel (**D**), *** *p* < 0.001 vs. CTR; ### *p* < 0.001 vs. GD. In panel (**E**), * *p* < 0.05 vs. CTR; *** *p* < 0.001 vs. CTR; ## *p* < 0.01 vs. GD; ### *p* < 0.001 vs. GD. Data are representative of at least three independent experiments (n = 3). Statistical analysis was performed by one-way ANOVA, followed by multiple comparisons performed by Bonferroni’s test.

**Figure 3 ijms-24-11303-f003:**
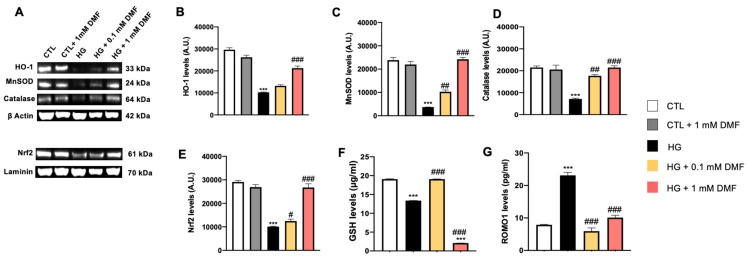
Effect of DMF on oxidative stress on SHSY5Y cells in high glucose. (**A**) Representative Western blots of proteins extracted from SHSY5Y cells treated as indicated above the blots. Blots were probed with the indicated antibodies. β-actin and laminin were used as loading controls. (**B**) Quantitative analyses of the blots revealed a significant decrease in HO-1 levels in the HG group compared to the CTL group. While 0.1 mM DMF did not affect HO-1 levels, 1 mM DMF prevented the HG-induced decrease in HO-1 levels. *** *p* < 0.001 vs. CTR; ### *p* < 0.001 vs. HG. (**C**) Quantitative analyses of the blots revealed a significant decrease in MnSOD levels following high glucose exposure. Notably, DMF prevented the decrease in MnSOD and increased MnSOD steady-state levels beyond the CTL group *** *p* < 0.001 vs. CTR; ## *p* < 0.01 vs. HG; ### *p* < 0.001 vs. HG. (**D**) Quantitative analyses of the blots revealed a significant decrease in catalase levels following high glucose exposure. Concentrations of 0.1 mM and 1 mM DMF significantly increased the expression of catalase compared to the HG group. *** *p* < 0.001 vs. CTR; ## *p* < 0.01 vs. HG; ### *p* < 0.001 vs. HG. (**E**) Quantitative analyses of the blots revealed a significant decrease in Nrf-2 levels following high glucose exposure. Concentrations of 0.1 mM and 1 mM DMF significantly increased the expression of Nrf-2 compared to HG group. *** *p* < 0.001 vs. CTR; # *p* < 0.05 vs. HG; ### *p* < 0.001 vs. HG. (**F**,**G**) GSH and ROMO1 levels in culture media were measured by ELISA. GSH levels were significantly reduced by high glucose exposure. A concentration of 0.1 mM DMF prevented such a decrease, however 1 mM DMF exacerbated the effects of high glucose on GSH levels. Glucose deprivation significantly increased ROMO1 levels; this increase was attenuated by DMF. In panel (**B**): *p* < 0.05 vs. CTR; ## *p* < 0.01 vs. HG. In panels (**C**–**E**): *** *p* < 0.001; ### *p* < 0.001 vs. HG. Data are representative of at least three independent experiments (n = 3). Statistical analysis was performed by One-way ANOVA, followed by multiple comparisons performed by Bonferroni’s test.

**Figure 4 ijms-24-11303-f004:**
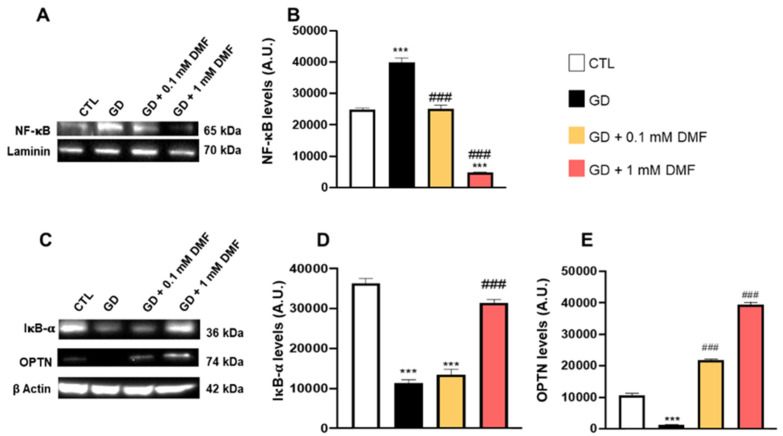
Effect of DMF on NF-κB in glucose deprivation conditions. (**A**) Representative Western blots of protein extracted from the nuclear fraction of SHSY5Y cells grown in low glucose. (**B**) Quantitative analysis of the NF-κB blots in the four groups tested. (**C**) Representative Western blots of proteins extracted from the cytosolic fraction of SHSY5Y cells and probed with the indicated antibodies. (**D**) Quantitative analysis of the IκB-α levels in the four groups tested. (**E**) Quantitative analysis of the OPTN levels in the four groups tested. Data are representative of at least three independent experiments. Statistical analysis was performed by one-way ANOVA, followed by multiple comparisons performed by Bonferroni’s test. In panel (**B**,**D**,**E**): *** *p* < 0.001 vs. CTR; ### *p* < 0.001 vs. GD.

**Figure 5 ijms-24-11303-f005:**
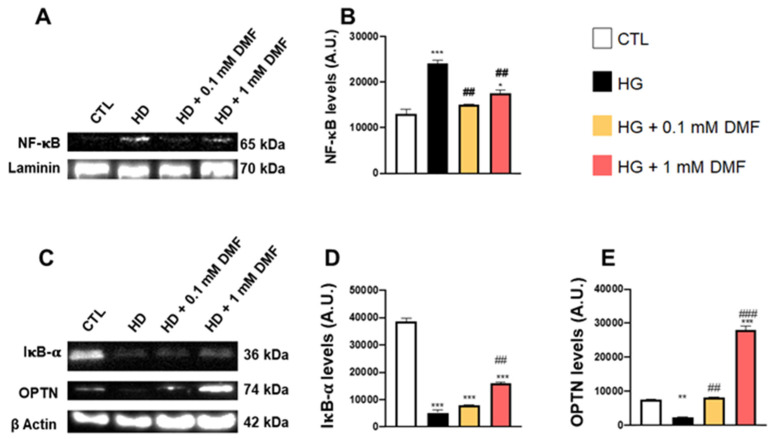
Effect of DMF on NF-κB in high glucose conditions. (**A**) Representative Western blots of proteins extracted from the nuclear fraction of SHSY5Y cells grown in high glucose. (**B**) Quantitative analysis of the NF-κB blots in the four groups tested. (**C**) Representative Western blots of proteins extracted from the cytosolic fraction of SHSY5Y cells and probed with the indicated antibodies. (**D**) Quantitative analysis of the IκB-α levels in the four groups tested. (**E**) Quantitative analysis of the OPTN levels in the four groups tested. Data are representative of at least three independent experiments. Statistical analysis was performed by one-way ANOVA, followed by multiple comparisons performed by Bonferroni’s test. For panel (**B**): * *p* < 0.05 vs. CTR; ** *p* < 0.01 vs. CTR; *** *p* < 0.001 vs. CTR; ## *p* < 0.01 vs. HG; For panels (**D**,**E**): *** *p* < 0.001 vs. CTR; ## *p* < 0.01 vs. HG; ### *p* < 0.001 vs. HG.

**Figure 6 ijms-24-11303-f006:**
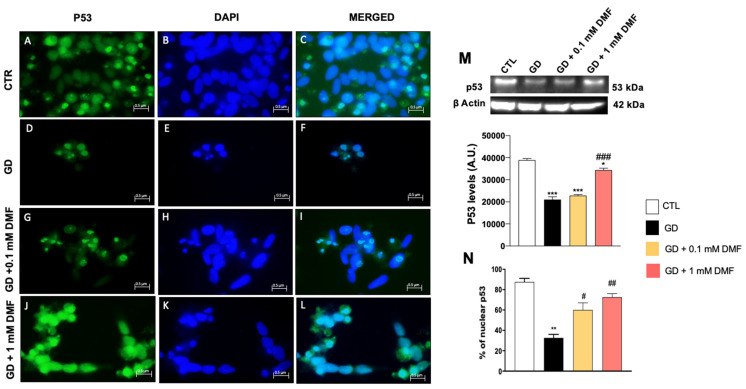
Effect of DMF on p53 expression on SHSY5Y cells in low glucose conditions. (**M**) Quantitative analyses of the blots revealed a significant decrease in p53 levels following glucose deprivation. A concentration of 1 mM DMF but not 0.1 mM DMF significantly increased the expression of p53 following glucose deprivation (**M**). *** *p* < 0.001 vs. CTR; * *p* < 0.05 vs. CTR; ### *p* < 0.001 vs. GD. Immunofluorescence revealed a significant decrease in p53 expression in the GD group (**D**–**F**,**N**) compared to the control group (**A**–**C**,**N**). Concentrations of 0.1 and 1 mM DMF restored p53 levels (**G**–**L**,**N**). Image magnification: 100×. Statistical analysis was performed by one-way ANOVA, followed by multiple comparisons performed by Bonferroni’s test. ** *p* < 0.01 vs. CTR; # *p* < 0.05 vs. GD; ## *p* < 0.01 vs. GD.

**Figure 7 ijms-24-11303-f007:**
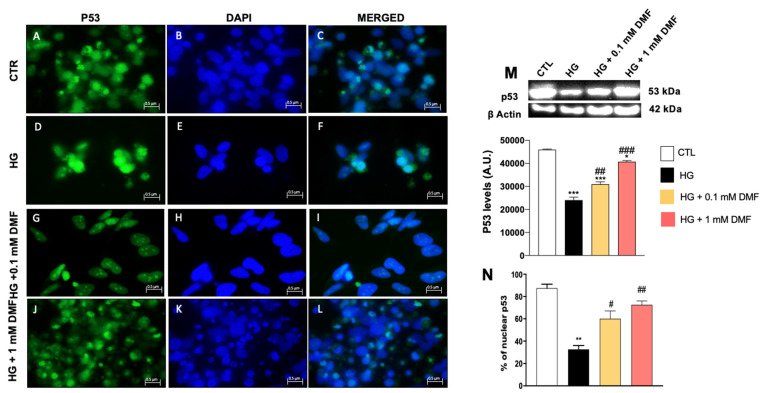
Effect of DMF on p53 expression on SHSY5Y cells in high glucose conditions. (**M**) Quantitative analyses of the blots revealed a significant decrease in p53 levels following high glucose exposure. Concentrations of 0.1 mM and 1 mM DMF significantly increased the expression of p53 compared to HG group (**M**). *** *p* < 0.001 vs. CTR; * *p* < 0.05 vs. CTR; ### *p* < 0.001 vs. HG; ## *p* < 0.01 vs. HG. Immunofluorescence revealed a notable decrease in p53 expression in the HG group (**D**–**F**,**N**) compared to the control group (**A**–**C**,**N**). Concentrations of 0.1 and 1 mM DMF rescued p53 expression (**G**–**L**,**N**). Image magnification: 100×. Statistical analysis was performed by one-way ANOVA, followed by multiple comparisons performed by Bonferroni’s test. ** *p* < 0.01 vs. CTR; # *p* < 0.05 vs. HG; ## *p* < 0.01 vs. HG.

**Figure 8 ijms-24-11303-f008:**
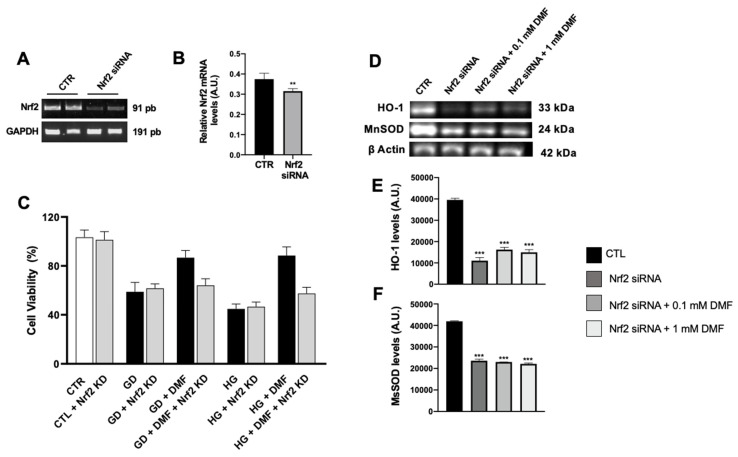
Effect of DMF in SHSY5Y following Nrf-2 siRNA silencing. (**A**) Nrf-2 mRNA expression in SH-SY5Y cells that were transfected for 48 h with either 20 nM control siRNA or 20 nM Nrf-2-specific siRNA was determined using real-time PCR. Values are normalized to GAPDH and expressed as fold change to untreated control cells. (**B**) The lack of Nrf-2 significantly reduced Nrf-2 mRNA expression compared to the control group, increasing SH-SY5Y susceptibility to metabolic damage. (**C**) DMF was not able to prevent cell death induced by GD or HG levels (64% and 57%, respectively). Data are representative of at least three independent experiments. ** *p* < 0.01 vs. CTR group. (**D**–**F**) Quantitative analyses of the blots revealed a significant decrease in HO-1 and MnSOD levels following Nrf-2 knockdown compared to CTR group. DMF was unable to increase the levels of these proteins in Nrf-2-knockdown cells. *** *p* < 0.001 vs. CTR. Statistical analysis was performed by one-way ANOVA followed by multiple comparisons performed by Bonferroni’s test.

**Table 1 ijms-24-11303-t001:** Description of primary antibodies used.

Antigen	Host Species	Type	Dilution	MW	Code	Source
HO-1	rabbit	polyclonal	1:500	33 kDa	sc-10789	Santa Cruz Biotechnology
MnSOD	rabbit	polyclonal	1:500	24 kDa	06-984	Merck Millipore
Catalase	mouse	monoclonal	1:500	64 kDa	sc-271803	Santa Cruz Biotechnology
Nrf-2	mouse	monoclonal	1:500	61 kDa	sc-365949	Santa Cruz Biotechnology
p53	mouse	monoclonal	1:500	53 kDa	sc-126	Santa Cruz Biotechnology
IκB-α	mouse	monoclonal	1:500	36 kDa	sc-1643	Santa Cruz Biotechnology
NFκB	mouse	monoclonal	1:500	65 kDa	sc-8008	Santa Cruz Biotechnology
Optineurin	mouse	monoclonal	1:500	74 kDa	sc-166578	Santa Cruz Biotechnology
β-actin	mouse	monoclonal	1:1000	42 kDa	sc-47778	Santa Cruz Biotechnology
Laminin A/C	mouse	monoclonal	1:500	70 kDa	sc-376248	Santa Cruz Biotechnology

## Data Availability

Data Availability could be requested to the authors.

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
