# Peer review of "An RNAi-Mediated Reduction in Transcription Factor Nrf-2 Blocks the Positive Effects of Dimethyl Fumarate on Metabolic Stress in Alzheimer’s Disease"

_ijms, 2023, doi:10.3390/ijms241411303_

Round 1

Reviewer 1 Report

the manuscript is interesting and generally well written. However, it presents some points that must be improved. In particular: 

Introduction: since NRf2 plays a key role in this manuscript, the multifaceted role of this transcription factor deserves to be highlighted. In fact, it plays an important role in preventing onset and progression of several types of cancer (see  PMID: 36641100 , 36335520,  35901941, 36289931 ). This is an important point to add since it can further highlight the interesting results found by the authors.

2.4. Western blot analysis: i suggest to insert the primary antibodies used in a dedicate table. 

Figures: the effect of the DMF treatment (alone) on HO1, MnSOD, Catalase and Nrf2 should be reported. Moreover, the number of replicates (N) should be inserted in the figure legends. 

Figure 7: higher magnifications must be shown. The current magnification is too low to appreciate p53 localization 

References: authors must follow the journal style

Author Response

Reviewer 1

The manuscript is interesting and generally well written. However, it presents some points that must be improved. In particular:

Introduction: since Nrf2 plays a key role in this manuscript, the multifaceted role of this transcription factor deserves to be highlighted. In fact, it plays an important role in preventing onset and progression of several types of cancer (see  PMID: 36641100 , 36335520,  35901941, 36289931). This is an important point to add since it can further highlight the interesting results found by the authors.

As suggested by the reviewer, the authors better described, in the Introduction section, the multifaceted role of Nrf2, to further highlight the interesting results found.

2.4. Western blot analysis: I suggest to insert the primary antibodies used in a dedicate table.

As suggested by the reviewer, the authors added a Table in the Materials and Methods section 2.4, also shown below, to better describe the primary antibodies used.

Antigen

Host species

Type

Dilution

MW

code

Source

HO-1

rabbit

polyclonal

1 : 500

33 kDa

sc-10789

Santa Cruz Biotechnology

MnSOD

rabbit

polyclonal

1 : 500

24 kDa

06-984

Merck Millipore

Catalase

mouse

monoclonal

1 : 500

64 kDa

sc-271803

Santa Cruz Biotechnology

Nrf2

mouse

monoclonal

1 : 500

61 kDa

sc-365949

Santa Cruz Biotechnology

P53

mouse

monoclonal

1 : 500

53 kDa

sc-126

Santa Cruz Biotechnology

IkB-?

mouse

monoclonal

1 : 500

36 kDa

sc-1643

Santa Cruz Biotechnology

NFkB p65

mouse

monoclonal

1 : 500

65 kDa

sc-8008

Santa Cruz Biotechnology

Optineurin

mouse

monoclonal

1 : 500

74 kDa

sc-166578

Santa Cruz Biotechnology

β-actin

mouse

monoclonal

1 : 1000

42 kDa

sc-47778

Santa Cruz Biotechnology

Laminin A/C

mouse

monoclonal

1 : 500

70 kDa

sc-376248

Santa Cruz Biotechnology

Figures: the effect of the DMF treatment (alone) on HO1, MnSOD, Catalase and Nrf2 should be reported. Moreover, the number of replicates (N) should be inserted in the figure legends.

As suggested by the reviewer, to evaluate the effect of DMF alone, the authors added DMF treatment alone in western blot analysis performed on HO1, MnSOD, Catalase and Nrf-2, in both DG and HG experiments, as shown in the new Figures 2 and 3. Moreover, as required, the number of replicates (n) was inserted in Figure legends.

Figure 7: higher magnifications must be shown. The current magnification is too low to appreciate p53 localization.

As suggested by the reviewer, the authors collected higher magnifications (100x) to better appreciate p53 immunofluorescence localization, as shown in the new Figures 6 and 7.

References: authors must follow the journal style.

As suggested by the reviewer, the authors revised the references according to the journal stile.

Reviewer 2 Report

It is a good paper with some relevance in the field. It can be of interest for the readers. Introduction is quite good, but they jump on to the AD pathology from the first line and the title is about general cell death? Maybe include AD in the the title?

Methodology is very well described.

Results section is good.

Same with the Discussion section. It suggest some nice correlations and mecanistics

Conclusions are necesary! no matter the format (rather strange) of the journal

English corrections also! I saw in some places terms such as "Thus far" etc

It is a good paper with some relevance in the field. It can be of interest for the readers. Introduction is quite good, but they jump on to the AD pathology from the first line and the title is about general cell death? Maybe include AD in the the title?

Methodology is very well described.

Results section is good.

Same with the Discussion section. It suggest some nice correlations and mecanistics

Conclusions are necesary! no matter the format (rather strange) of the journal

English corrections also! I saw in some places terms such as "Thus far" etc

Author Response

Reviewer 2

It is a good paper with some relevance in the field. It can be of interest for the readers. Introduction is quite good, but they jump on to the AD pathology from the first line and the title is about general cell death? Maybe include AD in the title?

As suggested by the reviewer, the authors changed the title to include AD pathology.

Methodology is very well described.

Thanks for your appreciation.

Results section is good. Same with the Discussion section. It suggest some nice correlations and mecanistics.

Thanks for your appreciation.

Conclusions are necessary! no matter the format (rather strange) of the journal

As suggested by the reviewer and according to journal style, the authors added a Conclusions section in the manuscript.

English corrections also! I saw in some places terms such as "Thus far" etc.

As suggested by reviewer, an English native speaker revised the manuscript to correct grammar and syntax errors

Round 2

Reviewer 1 Report

the manuscript has been significantly improved and can be accepted in the present form